# Development of Real-Time and Colorimetric Loop Mediated Isothermal Amplification Assay for Detection of *Xanthomonas gardneri*

**DOI:** 10.3390/microorganisms8091301

**Published:** 2020-08-26

**Authors:** Dagmar Stehlíková, Pavel Beran, Stephen P. Cohen, Vladislav Čurn

**Affiliations:** 1Biotechnological Centre, Faculty of Agriculture, University of South Bohemia, Na Sadkach 1780, 37005 Ceske Budejovice, Czech Republic; stehld00@zf.jcu.cz (D.S.); curn@zf.jcu.cz (V.Č.); 2Department of Plant Pathology, The Ohio State University, Columbus, OH 43210, USA; cohen.1107@osu.edu

**Keywords:** *Xanthomonas* spp., bacterial spot, LAMP

## Abstract

*Xanthomonas gardneri* is one of the causal agents of bacterial spot (BS), an economically important bacterial disease of tomato and pepper. Field-deployable and portable loop-mediated isothermal amplification (LAMP)-based instruments provide rapid and sensitive detection of plant pathogens. In order to rapidly and accurately identify and differentiate *X. gardneri* from other BS-causing *Xanthomonas* spp., we optimized a new real-time monitoring LAMP-based method targeting the *X. gardneri*-specific *hrpB* gene. Specificity and sensitivity of real-time and colorimetric LAMP assays were tested on the complex of bacterial strains pathogenic to tomato and pepper and on plants infected by the pathogen. The assay detection limit was 1 pg/μL of genomic DNA with an assay duration of only 30 min. The use of portable and handheld instruments allows for fast analysis, reducing the diagnosis time, and can contribute to proper disease management and control of *X. gardneri*. Due to the high efficiency of this method, we suggest its use as a standard diagnostic tool during phytosanitary controls.

## 1. Introduction

*Xanthomonas gardneri* is one of the species of the genus *Xanthomonas* that are currently considered to cause bacterial spot (BS) on tomato and pepper. It was first isolated in former Yugoslavia and named *Pseudomonas gardneri* [1]. Later, this name was suggested to be a synonym for *X. vesicatoria* [2]. However, DNA–DNA hybridization indicated that these organisms were genetically distinct [3]. Four distinct groups of *Xanthomonas* were differentiated based on various physiological and molecular tests [4]. Group A was named *X. euvesicatoria*; group B mostly included X. *vesicatoria*, group C *X. perforans* and group D *X. gardneri* [5]. Additionally, based on amplified fragment length polymorphism and multilocus sequence analysis (MLSA) of genes *atpD*, *dnaK*, *efp* and *gyrB* partial sequences, the four bacterial strains pathogenic for tomato and pepper were identified to the species level in *Xanthomonas* spp. [6]. The whole genomic sequence of *X. euvesicatoria* showed a 99.7% relationship with *X. perforans* [7]. It was suggested that the two species should be combined into one and subdivided into two pathovars: *X. euvesicatoria* pv. *euvesicatoria* and *X. euvesicatoria* pv. *perforans* [8]. Based on phylogenetic analysis and comparison of partial gyrase B gene sequences, *X. gardneri* and *X. cynarae* (an artichoke pathogen) were recognized as species closely related to *X. hortorum* [9]. New data from MLSA using four housekeeping genes (*dnaK*, *fyuA*, *gyrB* and *rpoD)* indicated that *X. gardneri* appears to be identical to *X. cynarae* but further investigation of the classification of *Xanthomonas* is still needed [10].

Various molecular, biochemical and physiological assays have been used to characterize *Xanthomonas* species. For *Xanthomonas* spp. causing BS, PCR-based molecular detection assays were used [11,12]. A region of *hrpB* operon as a potential source target for primers and probes for real-time TaqMan PCR assay for *X. euvesicatoria*, *X. vesicatoria*, *X. gardneri* and *X. perforans* was also evaluated [13]. PCR primer sets with TaqMan probes were developed to distinguish between *X. gardneri* and other *Xanthomonas* groups causing bacterial spot of tomato [14]. Currently, DNA-based techniques are the most used methods for molecular diagnostic of this group of pathogenic bacteria.

Nucleic acid amplification is one of the most valued methods in science but still requires sophisticated instruments for amplification and detection. A majority of the techniques for nucleic acid analysis utilize the polymerase chain reaction (PCR) amplification method, which requires repeated cycles of three temperature-dependent steps during the amplification of the target nucleic acid sequence [15,16]. Isothermal amplification techniques such as nucleic acid sequence-based amplification, helicase-dependent amplification, rolling circle amplification, multiple displacement amplification, recombinase polymerase amplification and loop-mediated isothermal amplification (LAMP) were developed to facilitate DNA amplification in simpler ways [17]. LAMP is a novel isothermal amplification method for amplifying a limited amount of DNA copies into millions of copies in less than an hour. It is a molecular technique of nucleic acid amplification where a set of four to six different primers binds to six to eight different regions on the target gene, providing high specificity [18]. A basic LAMP primer set consists of two outer primers (F3 and B3) and two inner primers (forward inner primer—FIP and backward inner primer—BIP). Loop primers may be used to accelerate the reaction [19]. LAMP methods utilize *Bacillus smithii*, *Bacillus stearothermophilus* or *Geobacillus* sp. DNA polymerases with high strand displacement activity at optimal working temperature ranged between 60 and 65 °C [18,20,21]. This method can be easily performed at the point-of-care thanks to its isothermal nature. It has been successfully used for rapid and specific detection of plant bacteria from infected plant tissues and soil. The objective of this study was to develop a real-time and colorimetric LAMP protocol for specific detection of *X. gardneri* that would be superior to the current PCR-based methods in its speed, ease of use and possibility of point-of-care application.

## 2. Materials and Methods

### 2.1. Bacterial Strains and Culturing

Bacterial strains (Table 1) were obtained from Belgian Co-ordinated Collections of Microorganisms—Bacteria Collection, Gent (BCCM/LMG); French Collection of Plant associated bacteria, Beaucouze Cedex (CFBP); Czech Collection of Microorganisms, Brno (CCM); Crop Research Institute Collection, Czech Republic, Prague—Ruzyně (CRI); Deutsche Sammlung Microorganismen und Zellkulturen GmbH, Germany (DSMZ); National Collection of Plant Pathogenic Bacteria, UK, York (NCPPB); National Collection of Agricultural and Industrial Microorganisms—(NCAIM); and Horticulture Research International, Wellesbourne, UK, (HRI-W). All *Clavibacter* spp. were grown on nutrient broth yeast extract (NBY) [22] at 27 °C for 3 to 7 days, depending on the subspecies. *Pantoea agglomerans* was grown on King’s B medium [23] at 25 °C for 24 to 48 h. Other bacteria were cultured on MPAg medium (meat–peptone agar with glucose: 20 g of nutrient agar no. 2, 2.6 g of yeast extract, 5 g of glucose, 10 g of agarose—added to 500 mL of distilled H_2_O, pH adjusted to 7.2 with 1 M NaOH and solidified) at 28 °C for 24–48 h, or 3–7 days (only *X. fragariae*).

### 2.2. DNA Isolation

Freshly grown bacteria (5–10 mg) were taken from plates. Total DNA was extracted using NucleoSpin^®^ Microbial DNA kit (Macherey-Nagel, Dylan, Germany) according to the manufacturer’s protocol. For agitation, a Retsch^®^ Mixer Mill MM400 was used for 4 min at maximal frequency (30 Hz). Concentration of extracted DNA was measured with a BioSpec-nano spectrophotometer (Shimadzu, Kyoto, Japan).

### 2.3. Primer Design

Partial sequence of the *hrpB* gene (GenBank ID: KX437681.1) was used for *X. gardneri* primer design [13]. One set of five primers (external primers F3 and B3; internal primers FIP and BIP and one loop primer LoopB) selected from total of three prospective primer sets was designed in PrimerExplorer software (Eiken Chemical Company, Tokyo, Japan; https://primerexplorer.jp/e/) and synthesized by Macrogen (Republic of Korea). Oligonucleotide sequences of the best primer set used for further testing are listed in Table 2. A sixth primer (LoopF) could not be designed due to the configuration of the template binding sites.

### 2.4. Real-Time LAMP

The LAMP reactions were performed in a QuantStudio™ 6 Flex Real-Time PCR System (Thermofisher, Waltham, MA, USA) and BioRanger^TM^ (Diagenetix, Honolulu, HI, USA). The reaction mixture contained 12.5 µL of Isothermal master mix (Optigene, Horsham, UK), 0.2 μM of each of outer primers (F3 and B3), 1.6 μM of each of inner primers (FIP and BIP) and 0.4 μM of loop primer (only LoopB). Lastly, 3 µL of template genomic DNA (10 ng/µL) was added and the reaction was brought to a final volume of 25 µL with nuclease-free H_2_O. The LAMP reaction mixtures were incubated for 60 min at 65 °C, followed by heating at 98 °C for 2 min to terminate the reactions. Melting analysis followed at temperature range from 70 °C to 99 °C with increments of 0.1 °C per second that allow for the generation of derivative melting curves. All LAMP assays were replicated at least three times, and all experiments included negative (no-template) controls.

### 2.5. Electrophoresis of LAMP

LAMP reaction products (5 and 10 µL, respectively) were analyzed via electrophoresis on a 1.5% agarose gel made of 1× TBE buffer (Tris/Borate/EDTA: 89 mM Tris, 89 mM boric acid, 2 mM EDTA, pH 8.3) stained with ethidium bromide (SigmaAldrich, St. Louis, MO, USA) at 90 V (4 V/cm) for 1 h. DirectLoad™ Wide Range DNA Marker was used for LAMP samples as a molecular standard for comparing molecular weight. To avoid contamination among samples, different laboratories were used for DNA extraction and for reaction mixture preparation, PCR Clean™ (Minerva Biolabs, Berlin, Germany, DE) was used for surface cleaning and only filtered pipette tips were used. After electrophoresis, the gel was visualized under UV illumination on GeneSys (Syngene, Cambridge, UK).

### 2.6. Colorimetric LAMP

Colorimetric LAMP reactions were carried out in a 25 μL volume containing 1.6 μM of each of inner primers FIP and BIP, 0.2 μM of each of outer primers F3 and B3, 0.4 μM of LoopB primer, 12.5 μL of 2× Colorimetric LAMP Master Mix (Cat. No. M1800, New England Biolabs) and 5 μL of DNA template (10 ng/µL). Reactions were incubated at 65 °C for 15 and 30 min in a heat block before results were recorded by naked eye.

### 2.7. Sensitivity and Specificity (Real-Time and Colorimetric LAMP)

Serial 10-fold dilutions of 10 ng/μL *X. gardneri* DNA ranging from 10 ng to 10 fg were used as a template for sensitivity test of DNA amplification assays. Specificity of assays was tested using *Xanthomonas* spp. strains and other related bacteria pathogenic for tomato and pepper (Table 1). No-template control (NC; water) was included in each LAMP run.

### 2.8. LAMP Assay on Plant Tissues

For testing of the LAMP assay with an infected plant tissue, 50 tomato and pepper plants were grown in growth chamber (Sanyo MLR-351H, Osaka, Japan) in temperature of 24 °C and 72% humidity for 4 weeks. Inoculation was done according to ISTA (2015) [24] methodology. Two youngest fully developed leaves were pierced in the area surrounding the major veins by six needles dipped in the suspension of *X. gardneri* type strain DSMZ 19127. First symptoms were observed after 3–4 weeks. Samples for subsequent testing were taken from the boundary of healthy and infected tissue and transferred to 50 µL of TE buffer. DNA extraction was performed as described above.

## 3. Results

In this study, we developed a LAMP-based real-time and colorimetric assay for specific, sensitive, reliable and robust detection and differentiation of *X. gardneri.* For primer design, we selected the *hrpB* gene due to its high species specificity.

LAMP assays distinctly amplified *X. gardneri* DNA, but none of the nontarget *Xanthomonas* species, including strains closely related to the *X. gardneri*, produced amplicons within 60 min (Figure 1a, Table 1). All the real-time LAMP assays were performed using the Isothermal master mix in both a QuantStudio™ 6 Flex Real-Time PCR System and a BioRanger^TM^ (Diagenetix, USA). Amplification of *X. gardneri* was first observable after 15 min of isothermal amplification. All reactions were subsequently analyzed on electrophoresis gel after 60 min of amplification, to confirm the results (Figure 1b). The *X. gardneri* reaction product displayed the typical ladder-like appearance of LAMP products, while no amplicons were detectable from the other reactions. Colorimetric LAMP assays were performed using 2× Colorimetric LAMP Master Mix in a heat block. No amplification was observed for any sample after 15 min. After 30 min, amplification was only visible in test tubes with *X. gardneri* DNA (Figure 1c).

To determine the sensitivity of real-time and colorimetric assays, genomic DNA diluted to concentrations ranging from 10 ng to 10 fg was used. A no-template control (water) was included in each experimental replication. The lowest concentration limit of detection was 1 pg/μL for *X. gardneri* (Figure 2a). However, very weak amplification was also observable via electrophoresis with concentrations of 100 fg and 10 fg (Figure 2b). Considering all the variants of the assay (real-time, electrophoresis, colorimetry), we defined the lowest amount of bacterial genomic DNA required to reliably detect the bacterium to 1 pg/μL after 30 min of amplification (Figure 2a–c) as an aggregate detection limit.

All *X. gardneri* infected tomato and pepper plant samples were positive for *X. gardneri*. No positive amplification was observed when LAMP primers were tested with healthy tomato plants (Figure 3a,b). The melting curves obtained for all *X. gardneri* isolates and from infected tomato plant showed the same melting temperature (T_m_) 94.27 °C, indicating similar sequences, and hence similar amplicons (Figure 3c). The possibility of using the LAMP assay in the field was tested on the BioRanger^TM^ platform using tomato plants inoculated by *X. gardneri* (Figure 3b). Amplification was detected by 30 min (45 min on BioRanger^TM^), suggesting that with our primers, this platform is a convenient and reliable method for point-of-care detection of *X. gardneri* if technological limitations of mobile devices are carefully considered. 

Although 30 min was enough to reliably detect the pathogen in most cases, we recommend 60 min of amplification due to possible variations in detection setup in different laboratories. The real-time nature of the LAMP assay may suggest its use for quantification of *X. gardneri* as well. However, our assay was not designed with quantification in mind, and all data resulting from following our recommendations should only be treated as detection, or not, of *X. gardneri*.

## 4. Discussion

Here, we developed a LAMP assay to detect *X. gardneri*, one of the causal agens of BS of tomato. LAMP assays are available for some other *Xanthomonas* spp. like *X. arboricola* pv. *pruni*, the causal agent of Stone Fruit Bacterial Spot; *X. oryzae* pv. *oryzae* the causal agent of Bacterial Blight disease; *X. oryzae* pv. *oryzicola*, the causal agent of Bacterial Leaf Streak disease; *X. campestris* pv. *musacearum*, the causal agent of Banana *Xanthomonas* Wilt; *X. citri* subsp. *citri* and *X. fuscans* subsp. *aurantifolii*, the causal agents of Citrus Bacterial Canker; and *X. translucens*, causal agent of cereal leaf streak [25,26,27,28,29]. LAMP assay for detection of *X. euvesicatoria*, the causal agent of BS of tomato, was also developed [30]. Our *X. gardneri*-specific assay substantially improves the detection of the bacteria associated with BS of tomato detection.

Different assays for detection of all the BS of tomato associated *Xanthomonas* were developed before. Most notably, many PCR based assays are available for single pathogens, and several multiplex-PCR assays for detection of all four species has also been developed [12,13,14,31,32]. In general, specificity of PCR based assays compared to LAMP seems to be similar. Sensitivity varies, and sometimes is not directly comparable (in some tests of sensitivity assessment bacterial cultures are diluted before DNA extraction, other tests use diluted DNA). With a detection limit 1 pg/μL of target DNA, our LAMP assay performs considerably better than the most sensitive *X gardneri* molecular detection assay by Araújo et al. [31] with detection limit of 50 pg/μL or 5 × 10^4^ CFU/mL.

The time to obtain results with LAMP is about 30 min, compared to one or more hours using real-time PCR or even more time-consuming end-point PCR. Although single LAMP reactions are usually slightly more expensive than PCR-based alternatives due to additional primers and more expensive polymerase, LAMP reactions can be performed on considerably cheaper and simpler devices such as the BioRanger^TM^ or Genie^®^ II (Optigene, Horsham, UK). This allows the use of LAMP assays in field conditions, which also makes LAMP more suitable for environments where high initial investment is not possible or desirable. Unlike PCR, LAMP does not directly provide information about size of amplified DNA, which is usually determined by performing melting analysis. For routine pathogen detection, this is not an issue, if the assay is well designed and tested. LAMP assays are, however, more difficult to design compared to PCR-based methods due to the higher complexity of the reaction and lower availability of software for LAMP assay design. Additionally, quantification is less accurate than real-time PCR [33]. Lower precision amplification devices (such as mobile devices like BioRanger^TM^), DNA extraction method, in-field contaminants and other factors may further decrease the precision of quantification, so we suggest only qualitative interpretation of results resulting from our assay. When preparing the assay, end users should carefully consider prolonging the time of amplification with mobile devices and for less efficient DNA extraction methods with possible contamination of sample DNA. For evaluation of the data, we suggest using the multi-operator validation test [30], which provides results without further statistical analysis and is therefore suitable for point-of-care detection.

The assay developed in this work is based on the gene sequence of *hrpB* gene. The *hrp* gene cluster was successfully used before as a target for molecular detection of phytopathogenic bacteria [13,34,35]. Other common targets include genes for ribosomal RNA, *atpD*, *gumD*, *gyrB*, *rpfB* and many others, or other non-annotated sequences obtained by whole genome analysis [28,36,37,38]. As expected from the literature, our own DNA alignment and BLAST search, the *hrpB* gene sequence of *X. gardneri* is conserved within the BS of tomato related *Xanthomonas* (with some variable regions within the group, allowing distinction of the four species), while being distinct enough from other related or tomato specific bacteria. This allowed us to design a LAMP primer set, which is specific only for *X. gardneri*, enabling improvement of current BS-related *Xanthomonas* detection and distinction between *X. gardneri* and other species. To date, no other LAMP assay for detection of this bacteria was published.

## Figures and Tables

**Figure 1 microorganisms-08-01301-f001:**
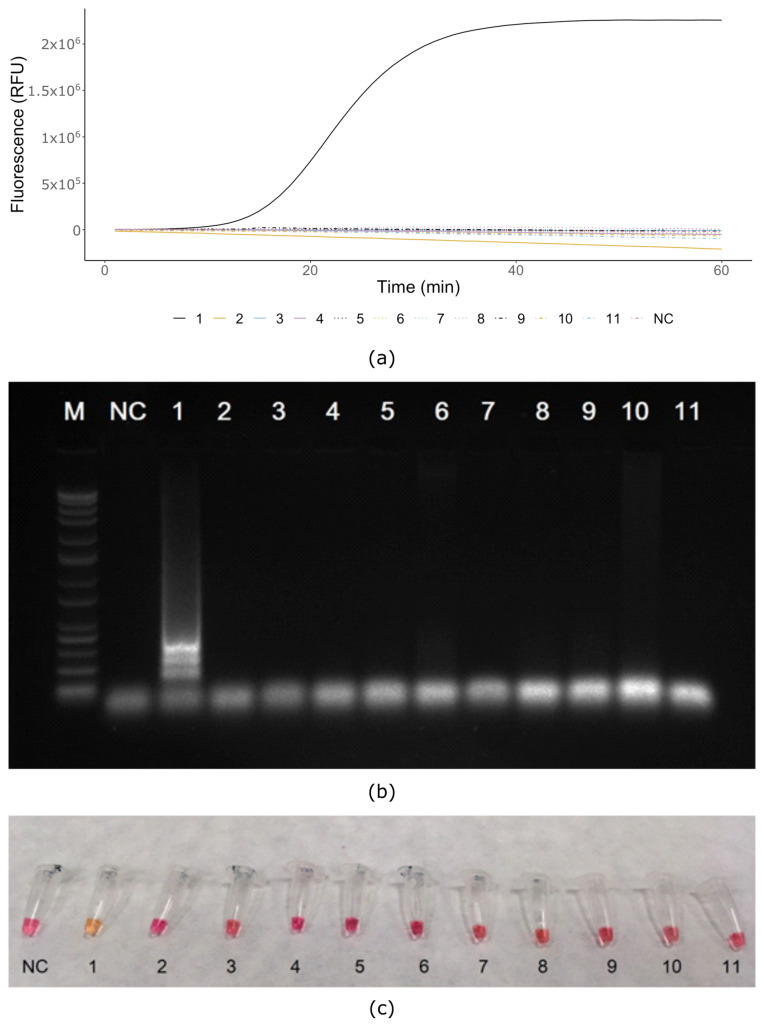
Specificity of selected bacterial species: (**a**) real-time LAMP assay (QuantStudio™ 6 Flex Real-Time PCR System); (**b**) gel electrophoresis of LAMP products confirms results of real-time LAMP; (**c**) Colorimetric Master Mix assay after 30 min reaction time; M—ladder 100 bp (NEB, Hitchin, UK), NC—negative control water, 1—*Xanthomonas gardneri* DSMZ 19127, 2—*Xanthomonas axonopodis* pv. *vesicatoria* CRI 1013, 3—*Xanthomonas perforans* DSMZ 18975, 4—*Xanthomonas vesicatoria* BCCM/LMG 920, 5—*Erwinia amylovora* CRI Ea10/96, 6—*Burkholderia glumae* BCCM/LMG 20138, 7—*Clavibacter michiganensis* susp. *michiganensis* CFBP 1460, 8—*Clavibacter michiganensis* subsp. *sepedonicus* NCPPB 3467, 9—*Pseudomonas syringae* pv. *syringae* NCPPB 2306, 10—*Pseudomonas syringae* pv. *tomato* CRI 8119, 11—*Ralstonia solanacearum* NCPPB 2505.

**Figure 2 microorganisms-08-01301-f002:**
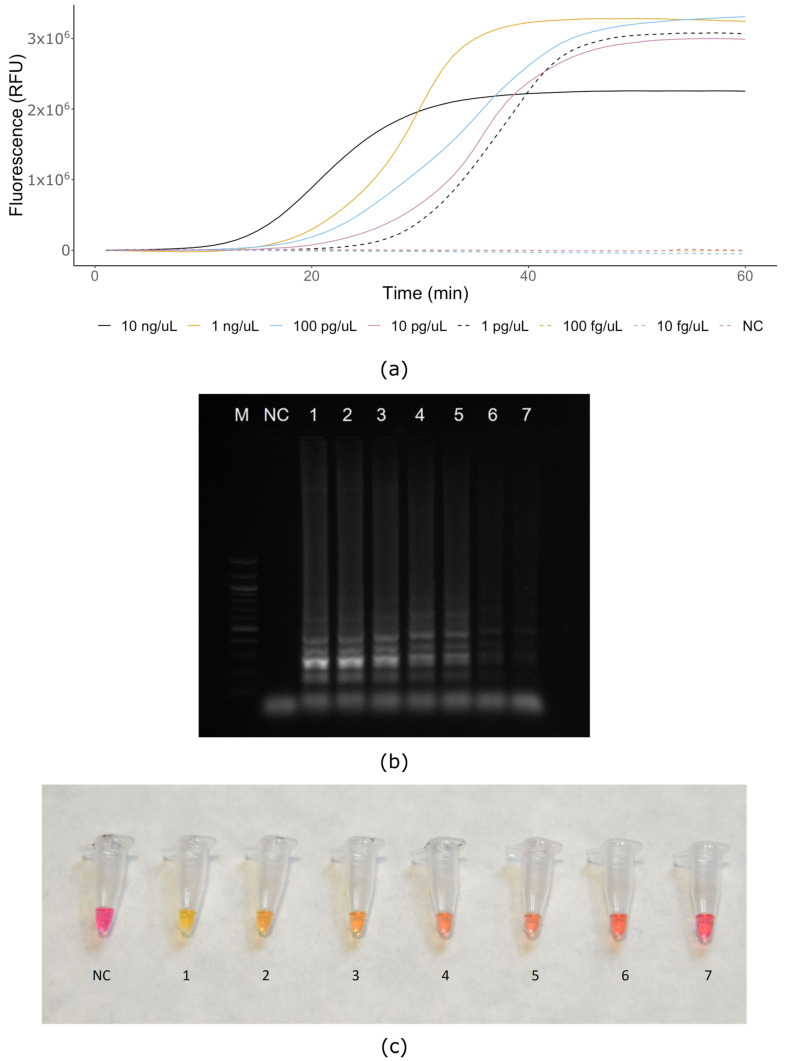
Serial 10-fold dilutions: (**a**) real-time LAMP assay (QuantStudio™ 6 Flex Real-Time PCR System); (**b**) gel electrophoresis of LAMP products confirms results of real-time LAMP; (**c**) Colorimetric Master Mix assay after 30 min reaction time; M—ladder 100 bp (NEB, UK), NC—negative control water, 1—10 ng/μL, 2—1 ng/μL, 3—100 pg/μL, 4—10 pg/μL, 5—1 pg/μL, 6—100 fg/μL, 7—10 fg/μL.

**Figure 3 microorganisms-08-01301-f003:**
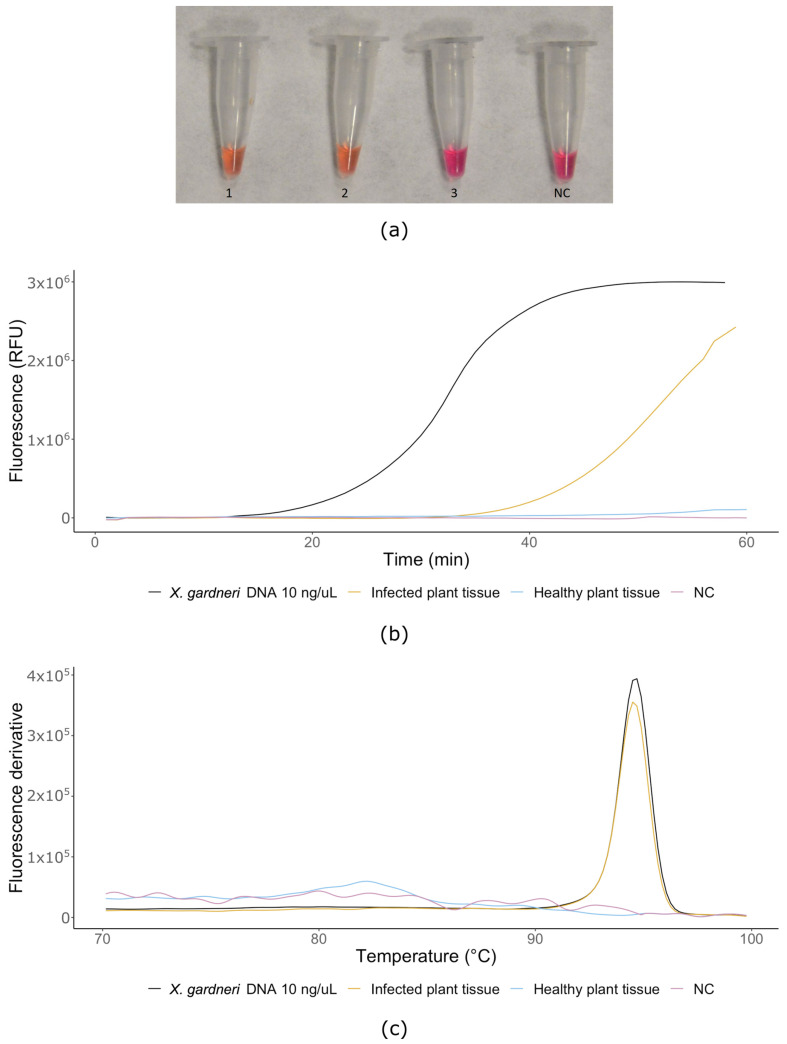
Naturally infected plants: (**a**) colorimetric LAMP assay, 1—*X. gardneri* bacterial culture DNA (10 ng/μL, strain DSMZ 19127), 2—infected plant sample, 3—healthy plant tissue, NC—non-template control (water); (**b**) real-time LAMP in BioRanger^TM^ device; (**c**) melting curves of LAMP products.

**Table 1 microorganisms-08-01301-t001:** ^1^ Bacteria listed in collection; ^2^ Type; ^3^ Pathotype; ^4^ not specified (N.S.) by collection; ^5^ (+) for positive detection, (−) for negative detection; BCCM/LMG—Belgian Co-ordinated Collections of Microorganismse Bacteria Collection; CFBP—French Collection of Plant Pathogenic Bacteria; CCM—Czech Collection of Microorganisms; CRI—Crop Research Institute Collection; NCPPB—National Collection of Plant Pathogenic Bacteria; DSMZ—Deutsche Sammlung von Mikroorganismen und Zellkulturen; NCAIM—National Collection of Agricultural and Industrial Microorganisms; HRI-W—Horticulture Research International, Wellesbourne, UK.

Name ^1^	Collection	Number in Collection	Geographic Origin	LAMP Result ^5^
*Xanthomonas gardneri*				
*X. gardneri* ^2^	DSMZ	19127	Yugoslavia	+
*X. gardneri*	CFBP	8588	France (Réunion)	+
*X. gardneri*	CFBP	7992	France (Réunion)	+
*X. gardneri*	CFBP	8120	Costa Rica	+
Other (non-*gardneri*) *Xanthomonas*				
*X. alfalfae* subsp. *alfalfae*	CFBP	3836	Sudan	−
*X. arboricola* pv. *pruni*	BCCM/LMG	854	New Zealand	−
*X. axonopodis* pv. *allii*	CFBP	6369	Not specified (N.S.) ^4^	−
*X. axonopodis* pv. *carotoae*	NCPPB	3440	Brazil	−
*X. axonopodis* pv. *vesicatoria*	CRI	1013	Czech Republic	−
*X. campestris* pv. *incanae*	HRI-W	6377	UK	−
*X. campestris* pv. *phaseoli*	NCAIM	B.01695	Hungary	−
*X. campestris* pv. *pisi*	NCAIM	B.01393	N.S. ^4^	−
*X. campestris* pv. *raphani*	HRI-W	8305	UK	−
*X. campestris pv. vesicatoria*	BCCM/LMG	934	Brazil	−
*X. campestris* pv. *vesicatoria*	BCCM/LMG	921	USA (Long Island)	−
*X. euvesicatoria*	BCCM/LMG	918	India	−
*X. euvesicatoria*	BCCM/LMG	922	USA (Florida)	−
*X. euvesicatoria*	BCCM/LMG	921	USA (Long Island)	−
*X. fragariae* ^2^	CFBP	6766	USA	−
*X. oryzae* pv. *Oryzicola* ^3^	CFBP	2286	N.S. ^4^	−
*X. perforans 9.2* ^2^	CFBP	7293	USA (Florida)	−
*X. perforans 9.2*	CFBP	8122	Thailand	−
*X. perforans* ^2^	DSMZ	18975	USA	−
*X. vesicatoria*	BCCM/LMG	925	Hungary	−
*X. vesicatoria* ^2^	CFBP	2537	New Zealand	−
*X. vesicatoria*	BCCM/LMG	920	Italy	−
Other species				
*Agrobacterium tumefaciens*	CCM	2835	Czech Republic	−
*Burkholderia glumae*	BCCM/LMG	20138	Philippines (province Jalajala Riza)	−
*B. glumae* ^2^	BCCM/LMG	2196	Japan (Ehime)	−
*Clavibacter michiganensis* subsp. *michiganensis*	CFBP	1460	France	−
*C. michiganensis* subsp. *sepedonicus*	NCPPB	3467	Poland	−
*Erwinia amylovora*	CRI	Ea10/96	Czech Republic	−
*Pantoea agglomerans* ^2^	CFBP	3845	N.S. ^4^	−
*Pseudomonas syringae* pv. *phaseolicola*	CRI	186/2	Czech Republic	−
*P. syringae* pv. *pisi*	NCPPB	3496	USA	−
*P. syringae* pv. *syringae*	NCPPB	2306	Italy	−
*P. syringae* pv. *tomato*	CRI	8119	Czech Republic	−
*Ralstonia pseudosolanacearum*	CFBP	3936	China (Guangdong)	−
*R. solanacearum*	NCPPB	2505	Sweden	−
*Stenotrophomonas* sp.	NCPPB	2859	Turkey	−

**Table 2 microorganisms-08-01301-t002:** Loop-mediated isothermal amplification (LAMP) primers.

Primer Name	Primer Length (nt)	Tm (°C)	Sequence (5′–3′)
F3	16	61.60	CGGGGTGCAGGTCAGC
B3	15	61.13	ACCGGCACCGCCAAG
FIP	37	-	CCACCTCGGCACGTTGCAGGCGAGGTATGCGAGTTGC
BIP	35	-	GCCGCCATCTCGCCTTGCGCCCCGATCCGATCACG
LB	17	61.26	CGAGCTGGTGGGCTTGT

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
