# Peer review of "Development of Real-Time and Colorimetric Loop Mediated Isothermal Amplification Assay for Detection of *Xanthomonas gardneri"

_microorganisms, 2020, doi:10.3390/microorganisms8091301_

Round 1
Reviewer 1 Report
While LAMP provides a rapid alternative to conventional PCR, or even real-time PCR, there are several issues and concerns for this manuscript that authors need to clarify.
- Authors did not identify the time to the threshold, and this will lead to a potential false positive (or negative) result, and will be difficult to be used for the real-world diagnostics.
- Figure 2, the highest concentration sample (10 ng/ml) had the lowest endpoint signal from the realtime LAMP, but not shown in the gel electrophoresis (panel B). This needs to be discussed.
- The time to the threshold from Fig. 2 A dose showed concentration-dependent, authors could plot the standard curve for the quantitation analysis and calculate/compare the bacterial load from the real-world sample.
- The authors claimed the detection limit is 1 pg genomic DNA. While the authors did not provide the time to the threshold, from Fig 2, it seems around 18 min at 1pg/ml, and around 12 min for 10 ng/ml. In Fig. 3, the time to the threshold for 10 ng/ml seems 15 min or longer, much longer than Fig. 2, and more importantly, for the naturally infected sample, the time to the threshold is over 33 min, which much longer than their detection limit, suggested this sample should be tested as negative. Indeed, in Fig.2, the run-time was only 30 min. Therefore, the method itself was not validated for sample analysis.
Other minor issues:
- In figure 2-a, authors could choose the colors that can be easily differentiated from one another or combining different colors and different line styles.
- Figure 3-b, the unit for time (x-axis) be consistent with Fig. 2. The three decimal in Fig 3b is not necessary.
Reviewer 2 Report
Stehlíková et al. reported the development of a LAMP-based real-time and colorimetric assay for detection and differentiation of X. gardneri, one of the causal agents of BS of tomato. They first designed their primers based on the hrpB gene and demonstrated that they targeted X. gardneri DNA with high specificity and produced amplicons within 30 min. Next they determined the sensitivity of this method by a serial genomic DNA dilution, and they found the detection limit was around 1 pg for X. gardneri. They finally applied this method for naturally infected tomato plants, and found that only X. gardneri infected tomato samples show positive response while not for healthy tomatoes. Overall, it could be a useful diagnostic tool for detection of X. gardneri. I have no other input and I would like to suggest publishing this manuscript as it is.
Author Response
Authors would like to thank the reviewer for the time spent by reviewing the manuscript and good overall evaluation of our work.
Reviewer 3 Report
Stehlíkova et al. describe the development of a LAMP assay for the detection of Xanthomonas Gardneri, a plant pathogen of tomato and pepper. Isothermal amplification of DNA portions of the pathogen hrpB gene allows to detect the infection by fluorescence measurement of the amplified products, later confirmed by gel electrophoresis analysis. Interestingly, the amplification can also be detected by naked eye, colorimetrically, allowing on-field testing of the plants. This is, in my opinion, the main advantage with respect to the usual PCR testing, even more than the shorter time necessary in the lab for proper read-out.
The paper is well conceived and written, allowing the reader to understand the logics beneath. The methods are also satisfactorily described. On the overall, I deem it fit for publication. There are, however, a few points that would need further clarification or discussion:
1) Colorimetric assay sensitivity: the authors claim that the method allows to detect by naked eye as little as 10 pg of DNA. However, it would be more correct to state that it can detect 10 pg/uL or the overall picograms that are necessary for the assay (I assume 50 pg if 10 pg/uL is the concentration of the 5 uL used for the test). Can the authors clarify this point?
2) The authors talk about LOD and LOQ of the fluorimetric assay quite qualitatively. In these cases, where the evalution is not purely visual, an actual quantification (LOD = 3S/N and LOQ = 10S/N, where S = signal and N = noise) should be provided.
3) The authors claim that the LOD and LOQ of PCR assays are difficult to compare, due to unequal treatment of samples. However, this does not impede to work out the concentration of DNA in the initial samples or anyway to go back to a common ground for assays comparison. This point is key to understand the real value of the assay proposed by the authors and they should thus better elaborate on this aspect, providing a comparison with existing methods.
4) Can the authors give more details on the criticalities in LAMP quantification mentioned in the conclusions?
Round 2
Reviewer 1 Report
In my opinion, there are still a couple of issues need to be clarified before acceptance.
- While authors provide rationale for not define the time to threshold, they need to discuss how they define the detection limit as 10 ng/ml. Have they tried lower concentration of DNA with 30 min run?
- It is bothering that authors used different machine to test the real sample (Fig. 2 vs Fig 3). It will provide more information that they use the cheaper machine (Bioranger) to run the standard samples, and compare with that from Quantstudio 6. If the detection limit is same or similar, then it provide a more economic way for the testing.
Reviewer 3 Report
The authors have carefully addressed all the points that I previously raised, adding comments, modifications and literature references where appropriate (e.g. sensitivity comparison with other published assays). Notably, they added numerous practical recommendations for the method end users, in order to improve reproducibility and ensure reliability of the results in practical applications. Personally, I appreciated this part, which is only seldom addressed in methodological papers.
For all these reasons, I believe that the manuscript is now ready for publication without further revision.
Author Response
Authors would like to thank the reviewer for the time spent by reviewing the manuscript, inspiring comments and good overall evaluation of our work.
Round 3
Reviewer 1 Report
A suggestion to authors that could clarified some confusions:
Mention the detection limit is based on using the QuantStudio6 system. When using BioRanger, the detection limit may get sacrificed. Indeed, by comparing Fig. 2 and 3, the 10 ng/mL sample in BioRanger system showed longer time-threshold (similar to 1 ng/ml sample on QauntStudio). Therefore, while BioRanger is more economic for field use, the detection limit may not match to that using QuantStudio. It would be helpful that authors use a couple sentences to discuss that point, so readers can understand the potential limitation for the BioRanger system.
